# Duckweeds for Phytoremediation of Polluted Water

**DOI:** 10.3390/plants12030589

**Published:** 2023-01-29

**Authors:** Yuzhen Zhou, Anton Stepanenko, Olena Kishchenko, Jianming Xu, Nikolai Borisjuk

**Affiliations:** 1School of Life Science, Huaiyin Normal University, Huai’an 223300, China; 2Leibniz Institute of Plant Genetics and Crop Plant Research (IPK), 06466 Gatersleben, Germany; 3Institute of Cell Biology and Genetic Engineering, National Academy of Sciences of Ukraine, 03143 Kyiv, Ukraine

**Keywords:** duckweed, *Spirodela*, *Lemna*, water pollutants, nitrogen, phosphorus, heavy metals, agrochemicals wastewater remediation

## Abstract

Tiny aquatic plants from the *Lemnaceae* family, commonly known as duckweeds, are often regarded as detrimental to the environment because of their ability to quickly populate and cover the surfaces of bodies of water. Due to their rapid vegetative propagation, duckweeds have one of the fastest growth rates among flowering plants and can accumulate large amounts of biomass in relatively short time periods. Due to the high yield of valuable biomass and ease of harvest, duckweeds can be used as feedstock for biofuels, animal feed, and other applications. Thanks to their efficient absorption of nitrogen- and phosphate-containing pollutants, duckweeds play an important role in the restorative ecology of water reservoirs. Moreover, compared to other species, duckweed species and ecotypes demonstrate exceptionally high adaptivity to a variety of environmental factors; indeed, duckweeds remove and convert many contaminants, such as nitrogen, into plant biomass. The global distribution of duckweeds and their tolerance of ammonia, heavy metals, other pollutants, and stresses are the major factors highlighting their potential for use in purifying agricultural, municipal, and some industrial wastewater. In summary, duckweeds are a powerful tool for bioremediation that can reduce anthropogenic pollution in aquatic ecosystems and prevent water eutrophication in a simple, inexpensive ecologically friendly way. Here we review the potential for using duckweeds in phytoremediation of several major water pollutants: mineral nitrogen and phosphorus, various organic chemicals, and heavy metals.

## 1. Introduction

Pollution and shortages of potable water are two of the most serious problems facing humanity. In many Asian countries and elsewhere, the demand for potable water doubles every 10–15 years due to rising domestic and industrial consumption [1,2]. In addition, eutrophication, the nutrient enrichment of municipal, agricultural, and industrial water reservoirs due to human activities leading to stimulation of bacteria, algae, and plant growth and oxygen limitation, is a global concern and has been identified as a major environmental problem for water resource management.

The need to reduce anthropogenic nutrients in aquatic ecosystems to prevent water eutrophication has been widely recognized [3], and a number of physical, chemical, and biological methods for wastewater treatment have been tested [4]. The cultivation of aquatic plants is an attractive option for restoring eutrophic water bodies, offering an eco-friendly method for removing nutrients, bioaccumulating toxic nutrients and heavy metals for disposal, and regulating oxygen balance [5]. Various aquatic plants have been used for the bioremediation of wastewater with varying degrees of success [6], with duckweeds standing out because of their specific physiology, high growth rates, multiple options for biomass usage, simple maintenance, and easy harvesting [7].

Plants bioremediate pollutants by diverse mechanisms, depending on the pollutant nature. Simple nutrients such as N and P, which result from agricultural runoff and cause the eutrophication of water sources, can be used as nutrients to fuel plant growth. Organic compounds can be detoxified by cellular metabolism within the plant or by associated microbes; for successful bioremediation, the plant must tolerate the doses of these organic compounds present in the environment, take up the compound, and be able to metabolize it [8]. Some pollutants, such as industrial dyes, can also be removed by biosorption, in which the pollutant binds to the surface of the plant. For pollutants that cannot be broken down, such as heavy metals, bioremediation may involve uptake and sequestration of the pollutant, followed by removal and processing of the contaminated biomass [9,10]. Understanding the mechanism of bioremediation has key implications for selecting the species used and improving the ability of that species to bioremediate the pollutant in question.

Duckweed is a common name that unites a group of floating aquatic plants in the *Lemnaceae* that inhabit all continents except Antarctica [11,12]. Because of their rapid propagation, among the fastest growth rates of flowering plants [13], duckweeds play an important role in the ecology of water reservoirs worldwide. Often seen as detrimental to the environment due to their ability to quickly colonize and take over bodies of water, duckweeds have a long history of applications in medicine, the food chain, and rituals since ancient times, from the Chinese Han dynasty, to early Christians, to classic Mayan culture [14]. Since the dawn of modern molecular biochemistry, duckweeds have served as a model plant helping to reveal basic functions of proteins, nucleic acids, and hormones, and have provided insights into plants development, photosynthesis, nutrient turnover, and other key processes in plants [15,16,17,18,19]. With the search for new sources of renewable energy and biomaterials, the 2010s saw duckweeds reemerge in academic research and a wide spectrum of new practical applications [20,21,22,23]. Duckweeds have well-recognized potential uses as animal feed, biofuel feedstock, and human food because of their rapid accumulation of biomass and its high protein and starch contents [20,24,25,26]. Here we present a comprehensive summary highlighting recent applications of duckweeds for phytoremediation of major water pollutants: (i) mineral nitrogen and phosphorus, (ii) various organic chemicals, and (iii) heavy metals.

## 2. Duckweeds (*Lemnaceae*): Tiny Aquatic Plants with Unique Properties

Although duckweeds are often mistaken for algae because of their small size and reduced morphology, phylogenetically they are ancient monocotyledonous flowering plants represented by 36 currently recognized species grouped into five genera: *Spirodela*, *Landoltia*, *Lemna*, *Wolffiella* and *Wolffia* (Figure 1). Compared to the majority of plant species, leaves, and stems in duckweeds are merged into a simplified structure known as a frond, and roots, which are entirely lacking in two genera (*Wolffia* and *Wolffiella*). Species of the genus *Spirodela* have the largest fronds, up to 15 mm across, while those of *Wolffia* species are 2 mm or less in diameter with *Lemna* species are of intermediate size at 6–8 mm.

Because of the ancient origin of duckweeds about 100 million years ago, their tiny size, and their simple morphology, the phylogenic grouping of this clade remains a matter of debate [12], and recently has become more dependent on new molecular methods [27,28,29]. For example, analysis of chloroplast and nuclear DNA markers supported renaming *Spirodela punctata* to *Landoltia punctata* and separating the species into a new genus *Landoltia* [30], as well as a recent reduction in genus *Lemna* from 13 to 12 species [31].

Due to their very rapid vegetative propagation, duckweeds can produce a biomass mat capable of covering expansive water surfaces and formed by a single species or different species. With a doubling time of about 24 h for some duckweed species, they are among the fastest-growing flowering plant known and can reach an annual biomass productivity of 39–105 tons of dry weight per hectare per year [13]. For comparison, the productivity of *Miscanthus*, a major grass used for bioenergy production, is 5–44 tons of dry weight per hectare per year. 

During biomass accumulation, duckweeds can very efficiently remediate different types of wastewater [32], and these traits can be further improved not only by selection of the best species or ecotypes [7,33,34], but also through optimization of plant’s growth parameters such as ration of nutrients, light intensity, fronds density, etc. [35,36,37,38]. Biomass accumulation by plants in general strongly depends on efficient use of N, and duckweed plants are extremely efficient at assimilating N. Due to N remobilization and recycling by duckweeds, their nitrogen use efficiency is extremely high, reaching more than 68 kg biomass/kg of N under N limitation [39]. Simultaneously, duckweeds demonstrate relatively high tolerance to many water pollutants (e.g., ammonia, heavy metals, various organic compounds) and other environmental stresses when used for remediation of agricultural, municipal, and even industrial wastewater streams. These complementary features of water remediation and fast biomass accumulation have made duckweed a subject of intense academic research interest from the business community in recent years [20,40].

Recently, there has been significant progress in the areas of duckweed genomics, biochemistry, and developmental physiology [41]. We now have up-to-date, fully sequenced genomes of two ecotypes of *Spirodela polyrhiza* [42,43], *Spirodela intermedia* W. Koch [44], *Lemna minor* L. [45], *Lemna minuta* [46] and *Wolffia australiana* (Benth.) Hartog and Plas [47], as well as ongoing whole-genome sequencing projects for *Lemna gibba* L. [48]. Those efforts are further supported by establishment of duckweed collections hosting more than 2000 ecotypes [49]. The major world duckweed depository is hosted by Prof. E. Lam at the Rutgers Duckweed Stock Cooperative at Rutgers University (New Jersey, USA). There are also a number of live in vitro collections available in Canada, China, Germany, Hungary, India, Ireland, and Switzerland [50]. The broad geographical distribution of these duckweed collections clearly reflects the worldwide interest in promoting duckweed research and its exciting new applications.

## 3. Duckweeds for Remediating Water Contaminated with Nitrogen and Phosphorus

Excessive use and runoff of agrochemical fertilizers, primarily those containing nitrogen (N) and phosphorus (P), are considered to be the major causes of eutrophication [51]. To maximize crop yields, about 80 million tons of nitrogen fertilizers are applied globally per year [52]. It is likely that no more than 40% of this amount is taken up by crops [53], and the rest eventually ends up in freshwater reservoirs en route to the ocean. Aquaculture systems are another serious source of water pollutants, and the pollution of water bodies by aquaculture has increased by 2–4% per year over the last 20 years in the Yangtze River Basin and Zhujiang Delta Basin of China [54]. The main contaminants from aquaculture wastewater effluent are ammonium, organic N, and P [55]. Only about 15% of the N and 25% of the P from the feed used in aquaculture are consumed by fish and shrimp, with the unused part accumulating in the water or sediment [54].

Duckweeds have potential uses for low-cost wastewater treatment and efficient removal of excess N and P [34,56,57,58]. It has been estimated that duckweed can accumulate up to 9.1 t/ha/year of total N and 0.8 t/ha/year of total P in their biomass [58]. It has been demonstrated that after just 3 days of incubation of the duckweed *Lemna turionefera* in local municipal wastewater, the main nutrient concentrations (total N and total P) were lower than those in the effluent from a local wastewater treatment plant. In the same study of Zhou et al. [58], within 15 days of growth, four duckweed species removed more than 93% of total N and total P in local municipal wastewater. The final total N concentration was 1 mg/L, which is much lower than the national standard for treated wastewater (15 mg/L, China Standard GB 18918-2002) and is close to the total N level accepted for drinking water (1.5 mg/L, China Standard GB3838-2002). Similarly, high rates of removal were also demonstrated with duckweed growing on sewage water [59] and wastewater from a hog farm [60]. Moreover, 98% removal of N and P from pig-farm effluent has been achieved [61]. This was accompanied by a significant increase in the level of dissolved oxygen and the production of duckweed biomass with 35% crude protein.

Another advantage of duckweed is its tolerance of relatively high levels of ammonium ion (NH_4_^+^), which can be toxic to plants, animals, and humans at high concentrations [62]. The common duckweed (*L. minor*) has been reported to grow well at NH_4_^+^ concentrations of up to 84 mg/L [63]. The ability of duckweeds to take up and tolerate such high levels of NH_4_^+^ makes them particularly suited to the remediation of wastewater from domestic, agricultural, and especially aquaculture sources, which often contain considerable amounts of NH_4_^+^ due to the breakdown of urea in urine and runoff of NH_4_^+^-containing fertilizers. Moreover, unlike most plant species, duckweeds prefer NH_4_^+^ over nitrate (NO_3_^−^) as the source of N, as first demonstrated for dotted duckweed (*Landoltia punctata*) [64,65], and more recently confirmed for five other duckweed species representing the genera *Spirodela*, *Lemna*, and *Wolffia* [66].

To optimizing nutrient and fertilizer use and promote plant productivity, much recent work has focused on plant–microbe interactions in the rhizosphere [67,68,69]. To attract and feed root-associated microbes, plants invest a substantial part of their photosynthesized carbon into rhizosphere exudates [70,71]. In terrestrial [69,72] and aquatic plants [33,73,74], microbial-mediated denitrification limits nitrogen assimilation by reducing nitrate and nitrite ions to volatile NO, N_2_O, and N_2_ [75]. However, denitrification might accelerate bioremediation of wastewater containing high levels of N compounds. A detailed analysis of denitrifying bacteria interacting with the common duckweeds *S. polyrhiza* and *L. minor* showed that some derivatives of fatty acids and stigmasterol, previously revealed as the major components of the cuticule in duckweed [76], participate in plant–microbe interactions stimulating bacterial nitrogen metabolism by activating nitrate and nitrite reductases [77,78]. Moreover, the denitrifying rhizospheric bacterium *Pseudomonas* sp. RWX31 was able to modulate the chemical composition of root exudates in duckweed, specifically inducing the secretion of stigmasterol. In turn, stigmasterol appeared to alter the composition of the rhizosphere microbial community in favor of denitrifying bacteria [77].

## 4. Duckweeds for Remediating Water Contaminated with Organic Compounds

With the continuous development of agriculture, industry and economy, more and more organic pollutants are generated from agricultural irrigation; chemical, pharmaceutical, papermaking and other industries; and domestic sewage. Some organic pollutants do not biodegrade well and tend to accumulate in the environment, endangering the food chain. They are also often teratogenic, carcinogenic, and/or mutagenic to animals and humans, and this seriously threatens ecological environment security and human health. Therefore, it is important to seek efficient, low-cost and sustainable technologies to remove organic pollutants from water. Below we summarized the studies on duckweeds interaction with a variety of organic pollutants in water, such as agricultural chemicals, pharmaceuticals, and personal care products (PPCPs) and other industrial organic compounds.

### 4.1. Organic Agrochemicals

With the increasing demand for food and the development of agriculture and aquaculture, tons of toxic agrochemicals such as pesticides, herbicides, and fungicides are produced and applied annually. A considerable amount of these chemicals applied on farmlands and aquaculture ends up in the aquatic environment without treatment, causing substantial pressure on the environment. Aquatic non-targeted organisms are more likely to be exposed to herbicides in multiple pulse events than long continuous exposure. Therefore, the potential of an organism to recover between exposures has important effects on the overall toxicity. In addition, the organism used for bioremediation must be able to tolerate relevant concentrations of the compound while taking up some of the compound to metabolize it. Studies to test the toxicity to and uptake of agrochemicals by duckweed have primarily used *L. minor*.

Most agrochemicals are tolerated by duckweed at low concentrations but are toxic at higher concentrations. Wilson and Koch (2012) evaluated the effects and potential recovery of *L. minor* exposed to the herbicide norflurazon for 10 days under controlled conditions [79]. Duckweed was severely inhibited by norflurazon, but there was a rapid recovery for all norflurazon concentrations tested after the plant was removed from the media [79]. Varga et al. (2020) evaluated the growth patterns and recovery potential of duckweed between multiple exposures to the herbicide isoproturon [80]. Growth was significantly inhibited during each exposure phase with significant cumulative effects in subsequent treatment cycles, resulting in a cumulative decrease in biomass production. However, inhibitory effects were reversible upon transferring plants to a herbicide-free nutrient solution. These results indicate that *L. minor* plants have a high potential for recovery even after multiple exposures to isoproturon.

Burns et al. (2015) investigated the ability of two duckweed species (*L. minor* and *L. gibba*) to recover from a 7-day exposure to different concentrations (0.4–208 µg/L) of the herbicide diuron [81]. Diuron significantly inhibited duckweed growth and biomass production after the initial 7-day exposure. Following transfer to herbicide-free media, recovery was observed for all effects at concentrations ranging 60–111 µg/L for *L. minor* and 60–208 µg/L for *L. gibba*. These results suggest that recovery is possible for primary producers at environmentally relevant concentrations that are considered significant in ecological risk assessment. The herbicide glyphosate can induce oxidative stress in plants through H_2_O_2_ formation by targeting the mitochondrial electron transport chain and the deleterious effects of the herbicide, glyphosate, on duckweed photosynthesis, respiration, and pigment concentrations were related to glyphosate-induced oxidative stress through H_2_O_2_ accumulation [82].

Even though agrochemicals are toxic to duckweed, researchers showed that duckweed was able to remove agrochemicals from the environment, indicating that this aquatic plant can efficiently eliminate organic contaminants and may ultimately serve as phytoremediation agents in the natural environment. Dosnon-Olette studied the effect of two herbicides, isoproturon and glyphosate, on *L. minor* growth [83] and showed that 10 µg/L isoproturon and 80 µg/L glyphosate had little effect on the growth rate and chlorophyll fluorescence of *L. minor*, which was able to remove 25% and 8% of the isoproturon and glyphosate, respectively, after a four-day incubation. Mitsou et al. (2006) studied the toxicity of the rice herbicide propanil to *L. minor* and found that propanil, at a concentration of 1 mg/L, did not affect the growth of *L. minor*, and did not induce antioxidative defenses within the plant. In addition, *L. minor* accumulated and metabolized the propanil [84]. Prasertsup and Ariyakanon (2011) explored the potential of water lettuce (*Pistia stratiotes* L.) and duckweed (*L. minor*) to remove different concentrations of the herbicide chlorpyrifos under greenhouse conditions. Low concentrations (0.1 and 0.5 mg/L) of chlorpyrifos had no significant effect on the growth of *L. minor* and *P. stratiotes*, but a higher concentration (1 mg/L) inhibited their growth. The maximum removal of chlorpyrifos (initial culture concentration of 0.5 mg/L) by *P. stratiotes* and *L. minor* was 82% and 87%, respectively [85]. Olette et al. (2008) compared the ability of three aquatic plants to remove three pesticides and found that compared to two other aquatic plants (*Elodea canadensis* and *Elodea canadensis*), *L. minor* more efficiently removed the pesticides, causing reductions of 50%, 11.5% and 42% for copper sulfate, dimethomorph, and flazasulfuron, respectively [86].

Many organisms limit toxicity of environmental factors by not taking up these factors; however, successful bioremediation requires that the plant take up some of the compound and metabolize it into less-toxic byproducts. Dosnon-Olette et al. (2010) studied the factors affecting the rate of pesticide uptake by two duckweed species, *L. minor* and *S. polyrhiza* [87]. Increased sensitivity to the pesticide dimethomorph was observed with increasing duckweed population density, possibly explained by having less light due to crowding. Plant photosynthesis uses light as the energy source leading to the production of biochemical energy (e.g., ATP) and reducing power (NADPH), which in turn are used for carbon fixation. This light-dependent electron source contributes to the absobing and transformation pesticide dimethomorph. Panfili et al. (2019) showed that *L. minor* is suitable for cleaning water polluted with the herbicide terbuthylazine, and this potential can be successfully improved by treating the species with a biostimulant or a safener such as Megafol and benoxacor [88].

Tront and Saunders (2007) evaluated the uptake and accumulation of a fluorinated analog of 2,4-dichlorophenol, 4-chloro-2-fluorophenol (4-Cl-2-FP), by *L. minor* [89]. Time series data gathered from an experiment with an initial aqueous-phase concentration of 130 mM 4-Cl-2-FP showed that 4-Cl-2-FP was continuously removed from the aqueous phase and less than 2% of original 4-Cl-2-FP was detected in plant tissue within the time period of 77 h. An increasing amount of metabolites was detected in the plant tissue, comprising 18.9%, 28.6%, and 53.4% of original 4-Cl-2-FP at 10 h, 24 h, and 77 h, respectively. This means that over 95% of the initial compound accumulated by duckweed was broken down in the plant cells. Although many studies have focused on herbicides and their effects on aquatic plants, other agrochemicals also affect plants. For example, Yılmaz and Taş (2021) examined the effect of the synthetic pyrethroid insecticide zeta-cypermethrin on the growth and bioremediation of aquatic photosynthetic organisms and showed that *L. minor* used zeta-cypermethrin as a nutrient and increased its development in low zeta-cypermethrin concentration (150 µg/L) medium [90]. However, high concentrations (300–600 µg/L) were toxic and inhibited growth. In addition, *L. minor* removed 35.4–95.9% of zeta-cypermethrin, depending on the initial concentration.

### 4.2. Pharmaceuticals and Personal Care Products (PPCPs)

PPCPs, including antibiotics, painkillers, anti-inflammatory drugs, disinfectants, and aromatics, pose potential hazards to the environment and human health. These pollutants are becoming ubiquitous in the environment because they cannot be effectively removed by conventional wastewater treatment due to their toxic and recalcitrant nature. Though the PPCPs, and particularly pharmaceuticals are usually present in wastewaters at very low concentrations of nanograms per liter, their average annual world per capita is 15 g with 50–150 g in the most developed countries [8,91]. Considering that these compounds are often pretty stable, bioactive and bioaccumulative, they can present serious environmental and human health risks [92,93].

Toxicity to plants caused by pharmaceuticals is an important issue, and several plant species, including duckweed, have been considered for phytoremediation of pharmaceuticals in wetlands [94]. Like agrochemicals, most PPCPs are toxic to duckweeds. For example, three β-blockers, propranolo, atenolol and metoprolol, were found to be toxic to duckweed (*L. minor*), which was less sensitive than the arthropod *Daphnia magna* and the green alga *Desmodesmus subspicatus* [95]. Kaza et al. (2007) evaluated the toxicity of 13 pharmaceuticals, usually at ng/L to µg/L in the environment, to duckweed *L. minor* [96]. A total of 7 out of 13 drugs tested were toxic at concentrations below 100 mg/L. The antipsychotic drugs thioridazine and chlorpromazine were the most toxic substances, having effective concentrations (EC_50_s) below 1 mg/L. Synthetic wastewater contaminated with the target compounds at 25 μg/L was prepared, and batch and continuous-flow experiments were conducted. Batch verification tests achieved removals of 98.8%, 96.4% and 95.4% for paracetamol, caffeine, and tricolsan, respectively. Overall removal of the PPCP contaminants was 97.7%, 98.0%, and 100% for paracetamol, caffeine, and tricolsan, respectively, by the constructed wetland system alone, while 97.5%, 98.2%, and 100%, respectively, were achieved by the lab-scale free water surface constructed wetland system [96].

In other work, Reinhold et al. (2010) tested the potential of both live and inactivated duckweed in removing pharmaceuticals in a microcosm wetland system [97]. Indeed, both live and inactivated duckweeds actively increased aqueous depletion of fluoxetine, ibuprofen, 2,4-dichlorophenoxyacetic acid, and the hand sanitizer triclosan. Some PPCPs can be used as a carbon source by duckweeds.

Amy-Sagers et al. (2017) conducted laboratory ecotoxicological assessments for a large range of concentrations of sucralose (an artificial sweetener) and fluoxetine (an antidepressant) on *L. minor* physiology and photosynthetic function [98]. Their results showed that, unlike humans who cannot break down and utilize sucralose, *L. minor* can use sucralose as a sugar substitute to increase its green leaf area and photosynthetic capacity. However, fluoxetine (323 nM) significantly decreased *L. minor* root growth, daily growth rate, and asexual reproduction.

#### 4.2.1. Antibiotics

Although most antibiotics are toxic to duckweeds, they can tolerate and phytoremediate those compounds from the environment with different efficiency depending on particular types and concentrations of the antibiotic. Cascone et al. (2004) evaluated the phytotoxicity of the fluoroquinolone antibiotic flumequine on *L. minor* and plant drug uptake [99]. Flumequine, at all concentrations between 50 and 1000 µg/L tested, affected plant growth, but duckweed continued to grow over a five-week period. In media containing flumequine, a large proportion of the drug (about 96% at all concentrations tested) was degraded in the presence of *Lemna*. Gomes et al. (2017) studied the mechanism by which PPCPs affect duckweeds and found that in *L. minor*, high concentrations of the common antibiotic ciprofloxacin disrupted the normal electron flow in the respiratory electron transport chain and induced hydrogen peroxide production, thus changing the photosynthetic, respiratory pathway, and oxidative stress capacity of duckweed and affecting its ability to remove ciprofloxacin [100]. Therefore, when the concentration of antibiotics is high, the metabolism of duckweed changes, affecting its ability to remove the antibiotics. Singh et al. (2018) evaluated the potential toxicity of the antibiotic amoxicillin on the duckweed *S. polyrhiza* and found it was toxic, even at low concentrations [101]. Nonetheless, the duckweed contributed directly to the degradation of antibiotics in the water. In other study, the same group [102] estimated the phytotoxicity and degradation by *S. polyrhiza* of the antibiotic ofloxacin. The high concentrations of ofloxacin caused a reduction in biomass (4.8–41.3%), relative root growth, protein (4.16–11.28%) and photopigment contents. The fronds treated with ofloxacin showed an increased level of antioxidative enzymes (catalas, ascorbate peroxidase and superoxide dismutase) than control. The residual ofloxacin content in the medium was significantly reduced (93.73–98.36%) by day seven and phytodegradation was suggested to be the main mechanism for removal of this antibiotic.

The specific mechanism of PPCP removal by duckweeds depends on the type of PPCP and the duckweed species. Iatrou et al. (2017) explored the mechanism of removal effect of four kinds of antibiotics by *L. minor* [103]. The removal efficiencies of *L. minor* were 100% (cefadroxil), 96% (metronidazole), 59% (trimethoprim), and 73% (sulfamethoxazole), respectively. Plant uptake and biodegradation were the major mechanisms accounting for metronidazole removal; the most important mechanism for trimethoprim was plant uptake.

#### 4.2.2. Analgesics and Anti-Inflammatory Drugs

The anti-inflammatory drugs and analgesics that do not require prescription in many countries, such as ibuprofen or paracetamol, are widely spread in the environment. Matamoros et al. (2012) found that caffeine and ibuprofen are removed by biodegradation and/or plant uptake by three aquatic plants, including the duckweed *L. minor*, and the removal rate was 83–99% in a microcosm wetland system [104]. In the presence of 1 mg/L ibuprofen, an increase in *L. gibba* frond number (+12%) and multiplication rate (+10%) was seen, while no variations in photosynthetic pigment content were observed [105]. Moreover, ibuprofen and 11 ibuprofen metabolites were detected in plants and in the growth medium, suggesting that *L. gibba* metabolizes ibuprofen. Li et al. (2017) studied the removal of four selected emerging PPCP compounds using greater duckweed (*S. polyrhiza)* in a laboratory-scale constructed wetland [106]. Di Baccio et al. (2017) explored the removal and metabolism of ibuprofen by *L. gibba* at high (0.20 and 1 mg/L) and environmentally relevant (0.02 mg/L) ibuprofen concentrations [107]. Ibuprofen uptake increased with increasing concentration, but the relative accumulation of ibuprofen and generation of hydroxy-ibuprofen was higher in the lower ibuprofen treatments. The main oxidized ibuprofen metabolites in humans (hydroxy-ibuprofen and carboxy-ibuprofen) were identified in the intact plants and in the growth solutions. Apart from a mean physical-chemical degradation of 8.2%, the ibuprofen removal by plants was highly efficient (89–92.5%) in all conditions tested.

### 4.3. Other Industrial Organic Compounds

Because of the efficient removal of pesticides and PPCPs by duckweed, researchers have explored whether duckweed can effectively remove other organic pollutants. In a recent study, the potential of *L. minor* for decolorization and degradation of malachite green (a triarylmethane dye) was investigated. The decolorization ability of the plant species was as high as 88%, and eight metabolic intermediate compounds were identified by gas chromatography-mass spectrometry [108]. Can-Terzi et al. (2021) [109] studied the phytoremediation potential of *L. minor* using methylene blue and showed that *L. minor* could effectively remove methylene blue from wastewater with the highest removal efficiency (98%) within 24 h. Fourier transform infrared spectroscopy (FTIR) and scanning electron microscopy (SEM) analyses indicated that dye removal was mainly by biosorption. Torbati (2019) evaluated the ability of *L. minor* to decolorize the acid Bordeaux B (ABB, an aminoazo benzene dye) [110]. Increased temperature and enhancement of initial plant weight increased the dye removal efficiency, but raising the initial dye concentration and pH reduced it. In optimum conditions, *L. minor* exhibited a considerable potential (94% removal) for the phytoremediation of ABB. Seven intermediate ABB degradation products were identified using gas chromatography-mass spectrometry analysis, indicating biodegradation is one of the mechanisms of L. minor’s removal and detoxification of ABB.

In a study of the fate of five benzotriazoles (used to inhibit the corrosion of copper) in a continuous-flow *L. minor* system, benzotriazole removal ranged between 26% and 72%. Plant uptake seemed to be the major mechanism governing the removal of benzotriazoles. When Zhang and Liang (2021) investigated the removal efficiency of 8 perfluoroalkyl acids by *L. minor* under aeration [111], they found that the removal efficiency of *L. minor* for long-chain perfluoroalkanes exceeded 95%, while the removal efficiency for short-chain perfluoroalkanes was marginal. The accumulation of perfluorooctane sulfonate in *L. minor* cells reached 14.4% after 2 weeks of exposure. Subsequently, the researchers further investigated the absorption and accumulation effect of *L. minor* on several intermediates of perfluoroalkyl compounds. The results showed that, after 14 days of exposure, *L. minor* accumulated 86.7 μG kg^−1^ and 1226 μG kg^−1^ for perfluorooctyl sulfonamide and fluorotelomere sulfonate, respectively [111]. In related work, Ekperusi et al. (2020) tested the potential of *Lemna paucicostata* (*Lemna aequinoctialis*, according to current classification) for removing petroleum hydrocarbons from crude oil-contaminated waters in a constructed wetland over a period of 120 days [112]. They found that *L. paucicostata* significantly (F = 253.405, *p* < 0.05) removed petroleum hydrocarbons from the wetland, reaching nearly 98% after 120 days, and estimated that about 97% of the petroleum hydrocarbons were biodegraded, because less than 1% bioaccumulated.

## 5. Duckweeds for Remediating Water Contaminated with Heavy Metals and Metalloids

### 5.1. Heavy Metals

Heavy metals are released into the environment from natural and anthropogenic sources, predominantly from mining and industrial activities. After entering the water environment, they accumulate in aquatic organisms, affecting their normal physiological and metabolic activities. Because they pose a threat to human health via the food chain and have serious impacts on the ecological environment, the removal of toxic pollutants is extremely important to minimize potential threats. Conventional techniques for the remediation of heavy metals are generally costly, time-consuming, and generate the problem of sludge disposal [113]. An environmentally friendly and economical treatment technology for the remediation of wastewater polluted with heavy metals is needed [114]. Duckweeds are relatively tolerant to heavy metals and able to take up many heavy metal ions, including those of cadmium, chromium, copper, iron, mercury, manganese, nickel, palladium, lead, and zinc [115,116,117,118,119,120,121,122,123,124]. Therefore, duckweed also has potential uses for monitoring and remediating heavy metals [125]. As a floating plant, duckweed can rapidly absorb heavy metals due to its special morphology and high growth rate [126]. In addition, duckweed can resist the toxicity of heavy metals through chelation and compartmentalization in vacuoles, effectively removing heavy metals in water through biological adsorption and intracellular accumulation [127].

A summary of studies of heavy metal uptake by duckweed species is shown in Appendix A. Different duckweed species have different tolerances to various heavy metals, and their biomass, photosynthetic pigments, and antioxidant enzyme activities are significantly different. The toxic effect of heavy metals on duckweed is the main factor limiting the application of duckweed. Therefore, identifying duckweed species that can tolerate specific heavy metals, have suitable bioaccumulation ability, and have suitable resistance will help to improve the phytoremediation of heavy metals in polluted water by duckweed.

Some researchers found that mixing different species of duckweed and coculturing duckweed with microorganisms or other plants can affect the absorption of heavy metals. Due to differences in tolerance and accumulation ability of different duckweed species for various heavy metals, the coculture of different duckweed species can improve both biomass and antioxidant enzyme activity, reducing the toxicity of heavy metals to duckweed and thus aiding the removal of heavy metals from polluted water [128]. By coculturing *L. punctata* and *L. minor* or individually in the medium with different concentrations of copper (Cu), Zhao (2015) found that coculturing produced better remediation effect than did single cultures at low Cu concentration; however, the single culture system was more effective at higher Cu concentration [129]. Duckweed can partly neutralize the toxic effect of high Cu concentrations by enhancing the activity of antioxidant enzymes, thus limiting the absorption of Cu.

The ability of duckweed to absorb heavy metals is also affected by the particular microorganisms symbiotically associated with the duckweed. Stout et al. (2010) showed that axenic duckweed, *L. minor*, accumulated slightly more Cd than did plants inoculated with bacterial isolates, suggesting that bacteria serve a phytoprotective role in their relationship with *L. minor* by preventing toxic Cd from entering plants [130].

Due to their ability to absorb heavy metals from the environment, duckweeds have been proposed for removing heavy metal contamination from wastewater. Bokhari et al. (2016) found that *L. minor* could effectively remediate both municipal and industrial wastewater [123], with removal rates of cadmium, copper, lead, and nickel all above 84% (Appendix A). In addition, because dried duckweed power has a large specific surface area and high porosity, duckweed can also be processed into dry powder and used as a potential new adsorbent. Chen et al. (2013) found that the lead ion (Pb^2+^) adsorption capacity of dried powder of *L. aequinoctialis* was more than 57 mg/g [131]. Nie et al. (2015) compared the removal rate of uranium ion (U^4+^) by live *L. puntata* and its dry powder and found that the removal rate of 5 g/L U^4+^ was nearly 96% by 1.25 g/L dry powder at pH 5, which is higher than that (79%) by 2.5 g/L (FW, fresh weight) live *L. puntata* [132]. Li et al. (2017) studied the adsorption of cadmium ion (Cd^2+^) in the aquatic environment by the dry powder of *S. polyrhiza* and *L. puntata* and found that the removal rates of Cd (50 mg/L) by the two kinds of dry powder of duckweed were 83% (*L. punctata*) and 96% (*S. polyrhiza*), respectively [133].

### 5.2. Metalloids: Boron and Arsenic

Boron (B) is an essential nutrient for plants but is toxic at high concentrations [134,135]. A study of the toxic effect of B (0.5–37 mg/L) on duckweed revealed that *S. polyrhiza* showed significantly reduced frond production and growth rates while significantly increasing the production of abnormal fronds. The authors concluded that *S. polyrhiza* could not remove significant amounts of B from the treatment solutions and, as a result, cannot be used as an effective component of B bioremediation systems [136]. Growing *L*. *gibba* at B concentrations of 0.3–10 mg/L showed no change in biomass production and a significant accumulation of B in fronds. At the same time, duckweed effectively reduced the B content in the environment in concentrations up to 2.0 mg/L [137]. A study of B toxicity using *L. minor* and *L. gibba*, with the aim of using them for phytoremediation and biomonitoring, revealed that significant inhibition of plant growth began at a B concentration of 16 mg/L. *L. minor* was more sensitive to B than *L. gibba*. The activity of the antioxidant enzymes superoxide dismutase, ascorbate peroxidase, and guaiacol peroxidase can serve as biomarkers for B-rich environments [138]. In another study, the combined use of *L. gibba* and chitosan beads effectively removed B from drinking water [139].

*L. gibba* showed the greatest potential to remove boron from irrigation water with B concentrations of 5.58–17.39 mg/L using a batch reactor. It was capable of removing 19–63% of the B from irrigation water, depending upon the level of contamination or initial concentration [140]. *L*. *gibba* and *L*. *minor* in the form of duckweed-based wastewater treatment systems coupled with microbial fuel cell reactor was shown to be an efficient method to simultaneously remove B from domestic wastewater/irrigation water and generate electricity [141,142]. In these studies, a monoculture of *L*. *gibba* showed the highest efficiency of B removal. Part of the research focused on the possibilities of B accumulation by duckweed under salt stress. Salt stress significantly affects the growth and B accumulation of L. minor. It was shown that only 7.9% to 15.5% of B was accumulated by L. minor during cultivation at NaCl concentration in a range of 0–200 mM. Finally, the authors concluded that L. minor is suitable for the accumulation of B when NaCl is below 100 mM [143]. Similar results were also shown for *S. polyrhiza* [144]. Thus, to date, information on the possibility of using duckweed for B removal is very limited, focusing on only three species, of which only *L*. *gibba* showed a sufficiently high potential for phytoremediation.

Arsenic (As) is present in the environment in inorganic and organic form and exists in four oxidation states—arsenate (As(V)), arsenite (As(III)), arsenic (As(0)), and arsine (As(-III)) [145]. Aquatic As phytoremediation approaches continue to be actively pursued [146,147]. Among 36 duckweed species, *L*. *gibba*, *L. minor*, *S. polyrhiza*, *W. globosa*, *W. australiana*, and *L. valdiviana* have been reported to remove As from water. The potential of duckweed for phytoremediation of As was first demonstrated in 2004 in waters from abandoned uranium mines. *L. gibba* revealed high arsenic bioaccumulation coefficients in wetlands of two former uranium mines in eastern Germany and under laboratory conditions. The potential extractions from mine surface waters using *L. gibba* were estimated to be 751.9 kg As/ha·yr [148]. In another study, *L. gibba* accumulated 10 times more As than background concentrations in the tailing waters of an abandoned uranium mine, reducing arsenic on average by 40.3% in the solutions [149].

*L. minor* showed high As accumulation (641 ± 21.3 nmol/g FW) when grown on As concentrations of 25–80 µM under laboratory conditions [150]. In another study, *L. minor* showed a removal rate of 140 mg As/ha·d, with a recovery of 5% when grown under a concentration of 0.15 mg/L [151]. The study of biological responses of *L. minor* revealed that both the duration of exposure and the concentration of inorganic As had a strong synergistic effect on antioxidant enzyme activity. *L. minor* showed a higher accumulation of As(III) compared to As(V) from polluted water [152]. A study of the accumulation of As by aquatic plants in running waters showed that *L. minor* is one of the top three studied species regarding arsenic accumulation (430 mg/kg DW). Higher values were observed only for *Callitriche lusitanica* and *Callitriche brutia* [153]. In hydroponics, *L. minor* revealed maximum removal of more than 70% As at a low concentration (0.5 mg/L) on day 15 of the experiment [154]. Another finding revealed that chelating agents had positive effects on As(III) or As(V) accumulation in *L. minor* [155].

For *L. valdiviana*, the As was only absorbed by the plant after a decline in the phosphate levels of the medium [156]. Concentrations greater than 1 mg/L As in the nutrient solution caused deleterious effects in *L. valdiviana* and compromised their phytoremediation capacity of water contaminated with As [156]. In addition, for *L. valdiviana*, As accumulation was dependent on pH. *L. valdiviana* accumulated 1190 mg/kg As (dry weight) from the aqueous media and reduced its initial concentration by 82% when cultivated between pH 6.3 and 7.0 [157].

At concentrations of 1.0, 2.0, and 4.0 µM As and dimethylarsinic acid, *S. polyrhiza* showed a significant level of As bioaccumulation, using different mechanisms for the degradation of arsenate vs. arsenite [158]. In addition, the uptake of inorganic arsenic (As (V) and As (III)) by *S. polyrhiza* was higher compared to the organic As sources, monomethylarsonic and dimethylarsinic acid. The addition of EDTA increased the uptake of inorganic As into the plant tissue, but the uptake of organic arsenic was not affected [159]. The study of the stability of *S. polyrhiza* at As (V) concentrations of 1, 5, 10, and 20 µM revealed an increase in the fresh biomass, photosynthetic pigments, and total protein contents of *S. polyrhiza* at lower concentrations of As (V) after 1 d of exposure, followed by a decrease in biomass with an increase in metal concentration [160]. In another study, *S. polyrhiza* showed the ability to survive in high concentrations of As (V) solution in hydroponics by decreasing As concentration, with a removal rate of 41% [161].

*W. globosa* accumulated 2–10 times more As than *S. polyrhiza/L. minor* and *Azolla* species [162]. At the low concentration range, the uptake rate was similar for arsenate and arsenite, but at the high concentration range, arsenite was taken up at a faster rate [162]. *W. globosa* was more resistant to external arsenate than arsenite but showed a similar degree of tolerance. A more detailed study of the mechanisms of As assimilation in *W. globosa* demonstrated an important role of phytochelatins in detoxifying As and enabling As accumulation [163]. A study conducted using *W. australiana* revealed the importance of microbial agglomerations for As assimilation. Sterile *W. australiana* did not oxidize As(III) in the growth medium or in plant tissue, whereas *W. australiana* with phyllosphere bacteria displayed substantial As(III) oxidation in the medium [164].

## 6. Conclusions and Perspectives

The beginning of the 21st century saw duckweed’s revival as a model system for academic research and a wide spectrum of new applications boosted by growing concerns related to wastewater, renewable energy sources, and rising fossil fuel prices [41]. Researchers and entrepreneurs regard duckweed as a powerful tool for bioremediation that can reduce anthropogenic pollution in aquatic ecosystems and prevent water eutrophication in a simple, cheap, and environmentally friendly way. This is clearly reflected in the number of PubMed publications related to the search terms “duckweed remediation” compared to other popular categories such as “duckweed feed”, “duckweed food”, or “duckweed biofuel” (Figure 2).

Testing different species and selecting duckweed varieties for more efficient phytoremediation remains an important and growing area of current research. Simultaneously, rapid advances in genome sequencing have revealed genes and metabolites responding to a growing number of specific pollutants and have provided valuable new information regarding the biochemical pathways underlying the uptake and assimilation of major pollutants by duckweed that are discussed in this review. This information, together with the optimized protocols for the genetic transformation of many duckweed species [165,166,167,168], should complement traditional selection efforts aimed at developing duckweed into an even more effective tool for protecting our environment and making our water as clean.

## Figures and Tables

**Figure 1 plants-12-00589-f001:**
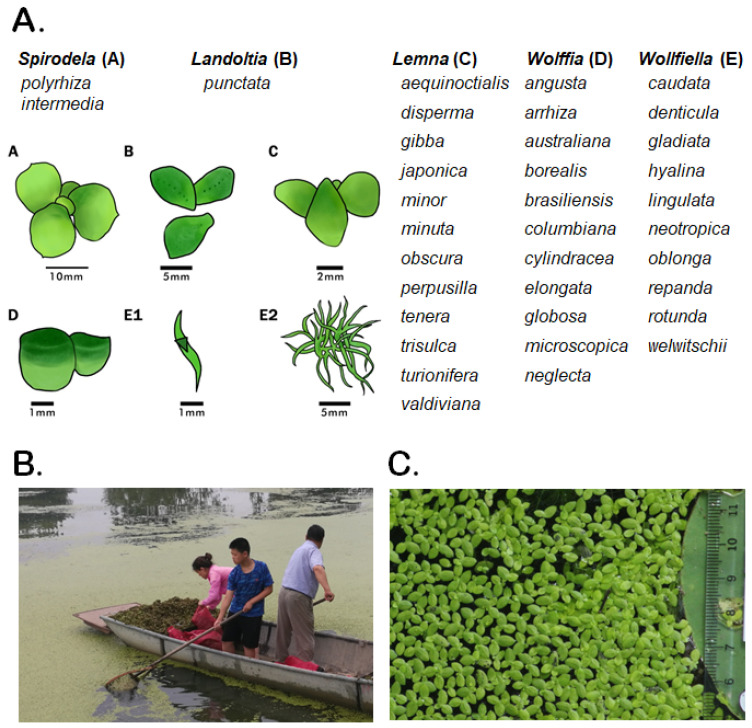
Duckweeds, the *Lemnaceae* plant family. (**A**) The *Lemnaceae* plant family contains 36 species grouped into five genera: *Spirodela*, *Landoltia*, *Lemna*, *Wolffia*, and *Wolffiella*. A, B, C, D, E1, E2 depict representative images of a species in the corresponding genera. The duckweed images are adapted from a drawing of Dr. K. Sowjanya Sree, Central University of Kerala, Periye, India [27]; (**B**) harvesting of duckweed covering a fishpond near Huai’an city, China; (**C**) *Lemna aequinoctialis* growing in the lake on the campus Huaiyin Normal University. Bar = 1 cm.

**Figure 2 plants-12-00589-f002:**
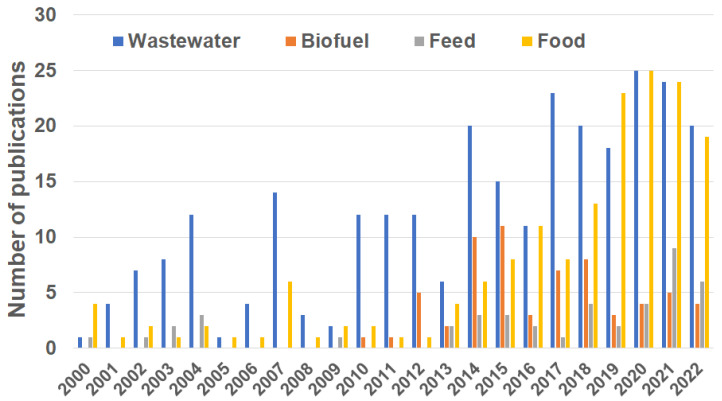
Publications related to duckweed research in the period 2000–2022. The total number of publications returned by the PubMed database (https://pubmed.ncbi.nlm.nih.gov, visited 3 November 2022) was 199 for “Wastewater”, 57 for “Biofuel”, 41 for “Feed”, and 147 for “Food”.

## Data Availability

Data are contained within the article.

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
