# Peer review of "Duckweeds for Phytoremediation of Polluted Water"

_plants, 2023, doi:10.3390/plants12030589_

Round 1

Reviewer 1 Report

The submitted paper entitled "Duckweeds for Phytoremediation of Polluted Water " is an interesting review/summary of the application of duckweeds in the phytoremediation of water pollution.
The paper can be published after modifications.
1. the paper is missing Table 1, which the authors describe on page 10.
2. on page 11 the authors mention the effect of the presence of NaCl on Lemna species. I think it is worth expanding this passage with more details.
3. Please review the paper and correct the spelling of Latin plant names.
4. p. 9 line 381-383 there is a sentence that is out of the context of the passage. Please remove this sentence.

Author Response

The paper can be published after modifications.
1. the paper is missing Table 1, which the authors describe on page 10.    

- Thanks for noticing. We meant Table S1, corrected.

  1. on page 11 the authors mention the effect of the presence of NaCl on Lemna species. I think it is worth expanding this passage with more details.

- The statement is complemented with the following extension: “Salt stress significantly affects the growth and B accumulation in L. minor. It was shown that only 7.9% to 15.5% of B was accumulated by L. minor during cultivation at NaCl concentration in range of 0-200 mM. Finally, authors concluded that L. minor is suitable for the accumulation of B when NaCl is below 100 mM [143]”.

  1. Please review the paper and correct the spelling of Latin plant names.

-Thanks for pointing out. The following corrections has been introduced:

Line 260 change “Lemna minor” to “L. minor

Line 261 change “Spirodela polyrhiza” to “S. polyrhiza

Line 366 change “Lemna gibba” to “L. gibba

Line 368 change “Lemna aequinoctialis” to “L. aequinoctialis

Line 371 change “Spirodela polyrhiza” to “S. polyrhiza

Line 547 change “S. polyrrhiza” to “S. polyrhiza”

  1. p. 9 line 381-383 there is a sentence that is out of the context of the passage. Please remove this sentence.

- The authors believe sentence line 381-383 “The decolorization ability of the plant species was as high as 88%, and eight metabolic intermediate compounds were identified by gas chromatography–mass spectrometry.” is in a right place. It actually explains the statement of the previous sentence, lines 379-381 “In a recent study, the potential of L. minor for decolorization and degradation of malachite green (a triarylmethane dye) was investigated”.

Reviewer 2 Report

Well written article, especially important due to the latest EU regulations on the reuse of treated wastewater in agriculture. Duckweeds should be considered during the modernization of small wastewater treatment plants.

Of course you paper needs some improvements.

My comments:

1. Keywords:  please add "phosphorus"

2.  Lines 98-100: You missed reference No [29], line 658-660

3. Lines 150 -15: In whose study?

4. Lines 223-471: there are much more information about "ability to recover" than about removal efficiency.  Please try to change the proportions.

5. Line 423: reference No [113] - I am not sure that thic apaper is abour heavy metals removal

6. Line 433: It should be Table S1

7. Line 549: in this brackets should be [163] 

Author Response

  1. Keywords:  please add "phosphorus"

 - Thank you for a good suggestion. Done

  1. Lines 98-100: You missed reference No [29], line 658-660  

- Reference [29] belongs to previous sentence.

  1. Lines 150 -15: In whose study?

- Explanation is added.

  1. Lines 223-471: there are much more information about "ability to recover" than about removal efficiency.  Please try to change the proportions.

  - Authors do not agree with this statement. The function “Find” marked the word “recover” 8 times compared to 46 times for “remov” in the indicated text section. In our opinion, this proportion is well in line with the main subject of the review. 

  1. Line 423: reference No [113] - I am not sure that this paper is about heavy metals removal

   - We thank the reviewer for noticing. The reference has been replaced with the correct one:  Dhaliwal S.S.; Singh J.; Taneja P.K.; Mandal A. Remediation techniques for removal of heavy metals from the soil contaminated through different sources: a review. Environ Sci Pollut Res Int 2020, 27(2, 1319-1333. doi: 10.1007/s11356-019-06967-1.   

  1. Line 433: It should be Table S1

Thanks for pointing out. Corrected.

  1. Line 549: in this brackets should be [163] 

Corrected. 

Round 2

Reviewer 1 Report

I have no comments.

It seems to me that the authors have made most of the changes.